# Unsupervised Object Representation Learning using Translation and Rotation Group Equivariant VAE

**Alireza Nasiri**
Simons Machine Learning Center
New York Structural Biology Center
anasiri@nysbc.org

**Tristan Bepler**
Simons Machine Learning Center
New York Structural Biology Center
tbepler@nysbc.org

## Abstract

In many imaging modalities, objects of interest can occur in a variety of locations and poses (i.e. are subject to translations and rotations in 2d or 3d), but the location and pose of an object does not change its semantics (i.e. the object's essence). That is, the specific location and rotation of an airplane in satellite imagery, or the 3d rotation of a chair in a natural image, or the rotation of a particle in a cryo-electron micrograph, do not change the intrinsic nature of those objects. Here, we consider the problem of learning semantic representations of objects that are invariant to pose and location in a fully unsupervised manner. We address shortcomings in previous approaches to this problem by introducing TARGET-VAE, a translation and rotation group-equivariant variational autoencoder framework. TARGET-VAE combines three core innovations: 1) a rotation and translation group-equivariant encoder architecture, 2) a structurally disentangled distribution over latent rotation, translation, and a rotation-translation-invariant semantic object representation, which are jointly inferred by the approximate inference network, and 3) a spatially equivariant generator network. In comprehensive experiments, we show that TARGET-VAE learns disentangled representations without supervision that significantly improve upon, and avoid the pathologies of, previous methods. When trained on images highly corrupted by rotation and translation, the semantic representations learned by TARGET-VAE are similar to those learned on consistently posed objects, dramatically improving clustering in the semantic latent space. Furthermore, TARGET-VAE is able to perform remarkably accurate unsupervised pose and location inference. We expect methods like TARGET-VAE will underpin future approaches for unsupervised object generation, pose prediction, and object detection. Our code is available at https://github.com/SMLC-NYSBC/TARGET-VAE.

## 1   Introduction

In many imaging modalities, objects of interest are arbitrarily located and oriented within the image frame. Examples include airplanes in satellite images, galaxies in astronomy images [1], and particles in single-particle cryo-electron microscopy (cryo-EM) micrographs [2]. However, neither the location nor the rotation of an object within an image frame changes the nature (i.e. the semantics) of the object itself. An airplane is an airplane regardless of where it is in the image, and different rotations of a particle in a cryo-EM micrograph are still projections of the same protein. Hence, there is great interest in learning semantic representations of objects that are invariant to their locations and poses.

In general, unsupervised representation learning methods do not recover representations that disentangle the semantics of an object from its location or its pose. Popular unsupervised deep learning methods for images, such as variational autoencoders (VAE) [3], usually use encoder-decoder frameworks in which an encoder network is learned to map an input image to an unstructured latent variable

36th Conference on Neural Information Processing Systems (NeurIPS 2022).

or distribution over latent variables which is then decoded back to the input image by the decoder network. However, the unstructured nature of the latent variables means that they do not separate into any specific and interpretable sources of variation. To achieve disentanglement, methods have been proposed that encourage independence between latent variables [4, 5]. However, these methods make no prior or structural assumptions about what these latent variables should encode, even when some sources of variation e.g. an object's location or pose are prevalent and well-known. Recently, several methods have proposed structured models that explicitly model rotation or translation within their generative networks [6, 7, 8] by formulating the image generative model as a function mapping coordinates in space to pixel values. Although promising, only the generative portion of these methods is equivariant to rotation and translation, and the inference networks have lackluster performance due to poor inductive bias for these structured latents.

To address this, we propose TARGET-VAE, a Translation and Rotation Group Equivariant Variational Auto-Encoder. TARGET-VAE is able to learn semantic object representations that are invariant to pose and location from images corrupted by these transformations, by structurally decomposing the image generative factors into semantic, rotation, and translation components. We perform approximate inference with a group equivariant convolutional neural network [9] and specially formulated approximate posterior distribution that allows us to disentangle the latent variables into a rotationally equivariant latent rotation, translationally equivariant latent translation, and rotation and translation invariant semantic latent variables. By combining this with a spatial generator network, our framework is completely invariant to rotation and translation, unsupervised, and fully differentiable; the model is trained end-to-end using only observed images. In comprehensive experiments, we show that TARGET-VAE accurately infers the rotation and translation of objects without supervision and learns high quality object representations even when training images are heavily confounded by rotation and translation. We then show that this framework can be used to map continuous variation in 2D images of proteins collected with cryo-EM.

## 2 Related Work

In the recent years, there has been significant progress in machine learning methods for unsupervised semantic analysis of images using VAEs [6, 10, 11, 12], flow-based methods [13], Generative Adversarial Networks (GAN) [14, 15, 16], and capsule networks [17]. These methods generally seek to learn a low-dimensional representation of each image in a dataset, or a distribution over this latent, by learning to reconstruct the image from the latent variable. The latent variable, then, must capture variation between images in the dataset in order for them to be accurately reconstructed. These latents can then be used as features for downstream analysis, as they capture image content. However, these representations must capture all information needed to reconstruct an image including common transformations that are not semantically meaningful. This often results in latent representations that group objects primarily by location, pose, or other nuisance variables, rather than type. One simple approach to address this is data augmentation. In this approach, additional images are created by applying a known set of nuisance transformations to the original training dataset. A network is then trained to be invariant to these transformations by penalizing the difference between low-dimensional representations of the same image with different nuisance transforms applied [18, 19]. However, this approach is data inefficient and, more importantly, does not guarantee invariance to the underlying nuisance transformations. This general approach has improved with recent contrastive learning methods [20, 21, 22], but these ultimately suffer from the same underlying problems.

Other methods approach this problem by applying constraints on the unstructured latent space [4, 5, 15]. In this general class of methods, the latent space is not explicitly structured to disentangle semantic latent variables from latents representing other transformations. Instead, the models need to be inspected post-hoc to determine if a meaningfully separated concepts have been encoded in each latent variable and what those concepts are. A few works have addressed explicitly disentangling the latent space into structured and unstructured variables. Sun et al. [17] propose a 3D point cloud representation of objects decomposed into capsules. Using a self-supervised learning scheme, their model learns transformation-invariant descriptors along with transformation-equivariant poses for each capsule, improving reconstruction and clustering. Bepler et al. [6] propose spatial-VAE, a VAE framework that divides the latent variables into unstructured, rotation, and translation components. However, only the generative part of spatial-VAE is equivariant to rotation and translation and the inference network struggles to perform meaningful inference on these transformations.

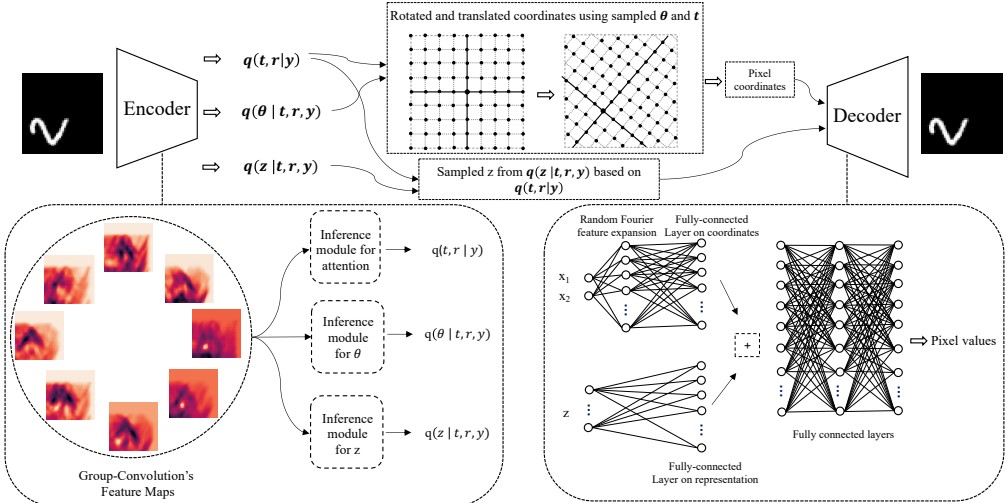

Figure 1: The TARGET-VAE framework. The encoder uses group-equivariant convolutional layers to output mixture distributions over semantic representations, rotation, and translation. The transformation-equivariant generator reconstructs the image based on the representation value $z$, and the transformed coordinates of the pixels.

To disentangle object semantic representations from spatial transformations, the inference network should also be equivariant to the spatial transformations. Translation-equivariance can be achieved by using convolutional layers in the inference model. However, convolutional layers are not rotation-equivariant. There has been many studies in designing transformation-equivariant and specifically rotation-equivariant neural networks [9, 23, 24, 25]. In Group-equivariant Convolutional Neural Networks (G-CNNs) [9], 2d rotation space is discretized and filters are applied over this rotation dimension in addition to the spatial dimensions of the image. This creates a network that is structurally equivariant to the discrete rotation group at the cost of additional compute. G-CNNs allow us to create a rotation-equivariant inference network.

A number of studies have proposed spatial transformation equivariant generative models, by modeling an image as a function that maps 2D or 3D spatial coordinates to the values of the pixels at those coordinates [7, 8, 26, 27, 6]. Due this mapping, any transformation of the spatial coordinates produces exactly the same transformation of the generated image. However, other than Bepler et al. [6], none of these studies perform inference on spatial transformations. To the best of our knowledge, TARGET-VAE is the first method to have rotation and translation equivariance in both the inference and the generative networks to achieve disentangling.

## 3    Method

TARGET-VAE disentangles image latent variables into an unstructured semantic vector, and rotation and translation latents (Figure 1). The approximate posterior distribution is factorized into translation, rotation, and content components where the joint distribution over translation and discrete rotation groups is defined by an attention map output by the equivariant encoder. The fine-grained rotation and content vectors are drawn from a mixture model where each of the joint translation and discrete rotation combinations are components of the mixture and are used to select mean and standard deviations of the continuous latents output by the encoder for each possible rotation and translation. Sampled translation, rotation, and content vectors are fed through the spatial decoder to generate images conditioned on the latent variables.

### 3.1    Image generation process

Images can be considered as the combination of discretely identifiable pixels, and the spatial transformation of an image is equivalent to transforming the spatial coordinates of its pixels. To have an image generation process adherent to the transformations identified in the latent space, we define our

generator as a function which maps the spatial coordinates of the pixels to their values. The generator outputs a probability distribution as $p(\hat{y}_i|z, x_i)$, where $z$, $x_i$, and $\hat{y}_i$ are the latent content vector, the spatial coordinate of the $i$th pixel, and the value of that pixel, respectively. Similar to [3], for an image with $n$ pixels, we can define the probability of the image generated in this manner as sum over probability of its pixel values:

$$log\ p(\hat{y}|z) = \sum_{i=1}^{n} log\ p(\hat{y}_i|z, x_i) \tag{1}$$

To define spatial coordinates over the pixels of the image, we use Cartesian coordinates with the origin at the center of the image. Translation and rotation of the image is achieved by shifting the origin and rotating coordinates around it, respectively. Depending on the value of the content vector, $z$, the generator acts as a function over the pixel coordinates to produce an image. In this setup, the generation process is equivariant to transformations of the coordinate system. Assuming $R(\theta)$ as the rotation matrix for angle $\theta$, and translation value of $t$, the probability of the generated image from Formula 1 is

$$log\ p(\hat{y}|z, \theta, t) = \sum_{i=1}^{n} log\ p(\hat{y}_i|z, R(\theta)x_i + t) \tag{2}$$

In this study, we focus specifically on rotation and translation of the spatial coordinates, but the image generative process extends to any transformation of the coordinate space.

## 3.2 Approximate inference on latent variables

We implement a VAE framework to perform approximate inference on content, rotation, and translation latent variables. Since rotation optimization is non-convex, we implement a mixture distribution over the 2D rotation space that allows the model to choose from $r$ discrete components. These components are used to approximate the posterior distributions over the rotation angle $\theta$. We represent the overall approximate posterior as $q(z, \theta, t, r|y)$, where $z$ is the latent content vector, $\theta$ is the rotation angle, $t$ and $r$ refer to the translation and discretized rotation components, and $y$ is the input image. By making the simplifying assumption that $q(z|t, r, y)$ and $q(\theta|t, r, y)$ are independent, the approximate posterior distribution factorizes as

$$q(z, \theta, t, r|y) = q(z|t, r, y)q(\theta|t, r, y)q(t, r|y), \tag{3}$$

where $q(t, r|y)$ is the joint distribution over discrete translations and rotations and $q(\theta|t, r, y)$ and $q(z|t, r, y)$ are the distributions over the real-valued rotation and the latent content vector conditioned on $t$, $r$, and the input $y$.

The Kullback-Leibler (KL) divergence between the approximate posterior and the prior over the latent variables is then

$$KL(q(z, \theta, t, r|y)||p(z, \theta, t, r)) = \sum_{z,\theta,t,r} q(z, \theta, t, r|y)log\frac{q(z, \theta, t, r|y)}{p(z, \theta, t, r)}. \tag{4}$$

To simplify this equation, we use the factorization from Equation 3 to reduce the KL-divergence to the following (see Appendix A1 for the full derivation), with the joint prior factorized into independent priors over the latent variables,

$$KL(q(z, \theta, t, r|y)||p(z, \theta, t, r)) = KL_{t,r} + \sum_{t,r} q\,(t, r)\,(KL_\theta + KL_z)\,, \text{where} \tag{5}$$

$$KL_{t,r} = \sum_{t,r} q(t, r|y)log\frac{q(t, r|y)}{p(t, r)},$$
$$KL_\theta = KL\,(q\,(\theta|t, r, y)\,||p\,(\theta|r))\,, \text{and}$$
$$KL_z = KL\,(q\,(z|t, r, y)\,||p\,(z))\,.$$

Multiplication of $Kl_\theta$ and $KL_z$ with $q(t, r)$ in Equation 5, weights the KL-divergence on $\theta$ and $z$ by the joint posterior distribution over $t$ and $r$. We choose to make the priors, $p(t)$ and $p(r)$, independent giving $p(t, r) = p(t)p(r)$, but this is straightforward to relax if a non-independent prior is desired.

We define the prior on $\theta$ to be independent of the translation and $z$ to be independent of the rotation and translation. We model $\theta$ as being drawn from a mixture distribution where the mixture components are the discrete rotations, $p(\theta) = \sum_r p(\theta|r)p(r)$. Given $r$, $\theta$ is drawn from a Gaussian distribution with mean given by the angle offset defined by $r$ and a fixed standard deviation which depends on the number of discrete rotation groups and is set to $\frac{\pi}{r}$. We tighten each of the $\theta$ mixture distributions by shrinking the standard deviation as $r$ grows to reduce overlap between these distributions when the number of discrete rotations is large. For the prior on $r$, we usually use a uniform distribution, but a discretized normal distribution can also be used to bias the model towards certain rotation angles if a non-uniform rotation distribution is known a priori. Then, $p(r)$ is calculated based on $\theta_{offset}$ for each discrete rotation. For example, when $r = 4$, then $\theta_{offset} \in \{0, \frac{\pi}{2}, \pi, \frac{3\pi}{2}\}$, which are the rotation angles of the discrete rotation group, $P_4$. The prior over translation, $t$, is Gaussian with mean 0, which denotes the center of the 2D coordinate system. The standard deviation of this prior depends on the number of pixels in our fixed-size coordinate system and whether we want the model to be flexible with large translation values or not. Lower standard deviation for the prior over translations, will more heavily penalize large predicted translation values.

The full variational lower-bound for our model is

$$\underset{(z, \theta, t) \sim q(z, \theta, t|y)}{\mathbb{E}}[log\ p(\hat{y}|z, \theta, t)] \quad - \quad KL(q(z, \theta, t, r|y)||p(z, \theta, r, t)), \tag{6}$$

where the KL-divergence term is defined in Equation 5.

### 3.3 Neural network architecture

#### 3.3.1 Group convolutional encoder

Convolutional layers are translation equivariant, meaning that a spatial shift in the image causes the same shift in the output feature map, but are not rotation-equivariant. To incorporate rotation-equivariance in the inference model, we use group convolutional layers [9]. A symmetry group $G$ is defined as a set $X$ with an operation ., which is associative on the set, and has inverse and identity elements. Furthermore, combination of two symmetry transformations on a set, also creates a symmetry group.

We define the $P_r$ group where $r \in \mathbb{N}$, and the operations consist of all translations, and $r$ discrete rotations, about any center of rotation in a 2D space. In the group convolutional layer, each kernel is rotated by $k\frac{2\pi}{r}$ angles, where $k \in \{0, 1, ..., r - 1\}$ and is then convolved with the input to produce an output map with $r$ values corresponding to the rotations of the kernels.

Following the group convolutional layer, we apply three 1x1 group convolutional layers (Figure 1), which are efficiently implemented as 3D convolutions. The final layer outputs the parameters of the approximate posterior distributions, $q(t = (i, j), r = r'|y) = \text{softmax}(a(y))_{i,j,r'}$ where $a(y)$ is an rxNxM activation map output by the network given input $y$, and $q(z|t = (i, j), r = r', y) = \mathcal{N}(\mu_z(y)_{i,j,r'}, \sigma_z(y)^2_{i,j,r'})$ and $q(\theta|t = (i, j), r = r', y) = \mathcal{N}(\mu_\theta(y)_{i,j,r'} + \frac{r'2\pi}{r}, \sigma_\theta(y)^2_{i,j,r'})$ where $\mu_z(y)_{i,j,r'}, \sigma_z(y)^2_{i,j,r'}, \mu_\theta(y)_{i,j,r'}$, and $\sigma_\theta(y)^2_{i,j,r'}$ are output by the inference network for each spatial location and discrete rotation group. $\frac{r'2\pi}{r}$ is the angle of the $r'$th discrete rotation.

During training, we sample from the joint posterior distribution of $q(t, r|y)$ using Gumbel-Softmax [28] in order to differentiably sample from the approximate posterior to estimate the expected reconstruction error given $q$. We use the soft one-hot sample to calculate the weighted sum of the parameters of the $q(z|t, r, y)$ and $q(\theta|t, r, y)$ distributions and then sample from those distributions to get a sample from the full joint approximate posterior. We define a coordinate system where $x$ is in the [-1,1] range, and is centered on the middle of the image. The output of the Gumbel-Softmax directly describes a sample from the approximate posterior over the translation, which we get by calculating the weighted sum over the translation grid points.

### 3.3.2 Spatial decoder

Given $t$ and $\theta$, we translate and rotate the image spatial coordinates, $x' = R(\theta)x + t$, where $R(\theta)$ is the rotation matrix for angle $\theta$. The generator receives the sampled $z$ value along with transformed spatial coordinates, $x'$, and outputs the value for the pixels at those spatial coordinates, forming the image. We use random Fourier feature expansion [29] on the coordinates of the pixel before feeding it through a stack of fully-connected layers (Figure 1). The spatial coordinate and $z$ inputs are processed separately by two parallel sets of fully-connected layers. Those representations are then summed and processed by the remaining shared fully-connected layers which give the final output value.

## 4 Results

### 4.1 Experiment setup

We run our experiments with $P_4$, $P_8$, and $P_{16}$ group convolutional layers in the inference model. The input images are normalized to the range of [0,1]. We use 128 rectangular kernels where size of the kernels in the first layer, is set to a value larger than the size of the targeted objects. The reason for this is that we only use one group convolutional layer in our models and we want to assure that its receptive field is at least as large as the objects of interest. We use leaky-ReLU activation functions, and the batch size for all the experiments is set to 100. We use ADAM optimizer [30], with learning rate of 2e-4, and the learning rate is decayed by the factor of 0.5 after no improvements in the loss for 10 epochs. We run the training for a maximum of 500 epochs with early-stopping in case of no improvements in the loss for 20 epochs. We run all our experiments in a single NVIDIA A100 GPU, with 80 GB memory.

Each fully-connected layer in the generator has 512 hidden units and the generator output is a probability value, in the case of the binary valued pixels, or mean and standard deviation for real valued pixels. For RGB images, the generator outputs R, G, and B values for each spatial location.

### 4.2 TARGET-VAE predicts the translation and rotation values of objects with high correlation

We first examine whether TARGET-VAE can accurately predict rotation and translation of objects despite being trained without supervision.

**Datasets** We use two variants of the MNIST dataset: 1) (MNIST(N)) MNIST digits randomly rotated by sampling from $\mathcal{N}(0, \frac{\pi^2}{16})$ and randomly translated by sampling from $\mathcal{N}(0, 5^2)$ pixels, and 2) (MNIST(U)) MNIST digits rotated by sampling uniformly from $U(0, 2\pi)$ and translated using the same distribution as in 1) (Appendix Figure 1). In both datasets, the train and test sets have 60,000, and 10,000 images of dimensions 50x50 pixels, respectively.

**Training** We train TARGET-VAE with $P_4$, $P_8$, and $P_{16}$ and z_dim=2 on these generated datasets. For the prior over $\theta$, we use $\mathcal{N}(0, \frac{\pi}{4})$ for the MNIST(N) dataset, and $U(0, 2\pi)$ for the MNIST(U). For $p(\theta|r)$ we calculate $p(\theta_{offset}) \sim p(\theta)$, where $\theta_{offset}$ are the rotation angles of the kernels in each dimension of r. For both datasets, we set the prior over $t$ to $\mathcal{N}(0, 5^2)$.

Given the trained model, we perform inference on $t$ and $\theta$ on the test set using the inference network. We find the most likely values of $t$ and $\theta$ from the approximate posterior distributions given by the network and compare these with the ground truth values for the test set images. We calculate the Pearson correlation between the predicted and true translations and the circular correlation between our predicted and true rotation angles [31]. In a comparison with a spatial-VAE network [6] trained with the same number of parameters as TARGET-VAE and z_dim=2, we find that TARGET-VAE significantly improves over spatial-VAE when predicting rotation. Spatial-VAE fails to give meaningful rotation predictions when digits are uniformly rotated, but TARGET-VAE predicts rotation with remarkable accuracy (Table 1). We also find that increasing the number of discrete rotations in the group convolution improves the accuracy of rotation inference, but does not have a significant impact on translation inference. Interestingly, spatial-VAE performs better than TARGET-VAE on translation inference, though the difference is small and predictions from both correlate strongly

with the ground truth. One possible reason for this is that TARGET-VAE only predicts whole pixel translations, due to the formulation of the approximate posterior, whereas spatial-VAE predicts real valued translations and, therefore, can predict sub-pixel values. In the future, including fine-grained translation inference in TARGET-VAE, similarly to the rotation distribution, could allow TARGET-VAE to resolve sub-pixel translations as well. We also found that increasing the dimension of $z$ did not have a noticeable effect on the ability of the network to predict translation and rotation.

Table 1: Translation and rotation correlation on MNIST(N) and MNIST(U) with z_dim=2

| | MNIST(N) | | MNIST(U) | |
| Group Convolution | Translation | Rotation | Translation | Rotation |
| --- | --- | --- | --- | --- |
| Spatial-VAE [6] | **0.994, 0.993** | 0.564 | **0.982, 0.983** | 0.005 |
| TARGET-VAE $P_4$ | 0.97, 0.972 | 0.814 | 0.975, 0.976 | 0.80 |
| TARGET-VAE $P_8$ | 0.981, 0.981 | 0.89 | 0.972, 0.971 | 0.859 |
| TARGET-VAE $P_{16}$ | 0.969, 0.972 | **0.898** | 0.974, 0.971 | **0.93** |

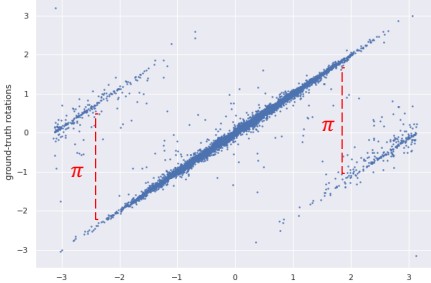 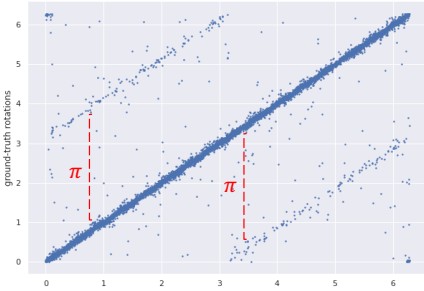

Figure 2: Left: Predicted $\theta$ vs. ground-truth data in MNIST(N), Right: Predicted $\theta$ vs. ground-truth data in MNIST(U), with TARGET-VAE $P_8$. There is a $\pi$ difference between the predicted angles and the ground-truth values of some of the digits (digits 0, 1, and 8), which is related to the rotation symmetry of them.

Some of the digits in MNIST have natural ambiguity in their rotations. Investigating the correlation between the predicted rotation values and the ground truth ones shows a difference about $\pi$ between some of the predicted angles and the ground truth (Figure 2). This is expected, because several digits have approximate symmetry. For example, 0, 1, and 8 are roughly 2-fold symmetric (Appendix A.3) creating inherent ambiguity in the rotation that is reflected in larger error in the rotation predictions (Appendix Table 1). Six and 9 are also similar when rotated by $\pi$, although, interestingly, TARGET-VAE is able to distinguish these digits and predict their rotations accurately. Perhaps this is because 9 is typical drawn with a straight spine while 6 is drawn with a curved one. This also suggests that the posterior predictions given by TARGET-VAE could be helpful for identifying the symmetry group of objects.

### 4.3 TARGET-VAE improves clustering accuracy by learning translation and rotation invariant features

By disentangling rotation and translation from the representation values, TARGET-VAE is able to recover semantic organization of objects. Using the z_dim=2 models trained above and additional TARGET-VAE models trained with z_dim=32, we extract the most likely value of $z$ for each model from $q(z|t, r, y)$ for each image in the MNIST(N) and MNIST(U) test sets. To evaluate the disentangled semantic value of these representations, we cluster the test set images in $z$-space using agglomerative clustering [32]. We then calculate the clustering accuracy by finding the highest accuracy assignment between clusters and digits for each set of representations. We find that semantic representations produced by TARGET-VAE much more closely represent the underlying semantics of the MNIST dataset when compared to representations produced by baseline methods (Table 2). TARGET-VAE achieves clustering accuracies approximately 2-5x higher than standard VAE [3], beta-VAE [4], and spatial-VAE [6] models trained with the same number of parameters as TARGET-

Table 2: Clustering accuracy (%) on MNIST(N) and MNIST(U)

| Model | MNIST(N) | MNIST(U) |
|---|---|---|
| VAE (z_dim=4) [3] | 15.3 | 12.8 |
| Beta-VAE (z_dim=4) [4] | 15.1 | 18.0 |
| Spatial-VAE (z_dim=2) [6] | 37.1 | 28.2 |
| TARGET-VAE $P_4$ (z_dim=2) | 56.4 | 56.6 |
| TARGET-VAE $P_8$ (z_dim=2) | 60.1 | 57.1 |
| TARGET-VAE $P_{16}$ (z_dim=2) | 60.1 | 63.4 |
| TARGET-VAE $P_4$ (z_dim=32) | 65.1 | 64.3 |
| TARGET-VAE $P_8$ (z_dim=32) | **77.7** | 69.1 |
| TARGET-VAE $P_{16}$ (z_dim=32) | 75.2 | **71.2** |

VAE. We set the z_dim=2 for both spatial-VAE and TARGET-VAE. Since the standard VAE and the beta-VAE do not have dedicated latent variables for identifying translation and rotation, we set the dimension of the unstructured latent space to 4 for these models. Also, we use $\beta = 4$ for the beta-VAE based on the settings proposed by Higgins et al. [4]. We believe that direct inference on rotation and translation with the equivariant inference network of TARGET-VAE plus the transformation-equivariant generator are the main reason for the dramatic increase in the clustering accuracy (see ablation studies in Appendix A.7).

We also observe that increasing the dimension of $z$ can significantly increase clustering performance and produces clearly distinct clusters for each digit (Figure 3). Increasing the number of discrete rotations also can improve clustering performance, likely because improving rotation inference means that $z$ and $\theta$ are better disentangled, but this trend is not universally true for both MNIST(N) and MNIST(U). We note that the $P_8$ model with z_dim=32 performs best on MNIST(N), perhaps because rotation inference is performed roughly equally well by the $P_8$ and $P_{16}$ models on this dataset.

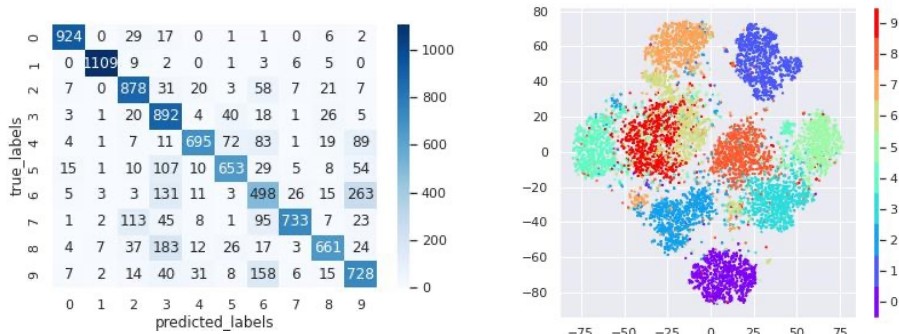

Figure 3: Confusion matrix of trained TARGET-VAE on MNIST(N) with z_dim=32 (left), and the t-sne figure of the 32 dimension latent representation space (right). Based on the confusion matrix, most of the digits are clustered correctly, and the largest misclassification is related to the confusion between rotated digits 6 and 9, which is understandable.

TARGET-VAE is also able to detect multiple objects at inference time (Appendix A.4) and is able to learn meaningfully disentangled semantic representations on other datasets, including dSprites [33] (Appendix A.5) and the galaxy zoo [1] (Appendix A.6).

## 4.4 Identifying class-averages and protein conformations in the cryo-EM micrographs with TARGET-VAE

In single-particle cryo-EM micrographs, individual biomolecules (particles) are spread throughout images with random orientations and locations. Being 2d projections of 3d structures, particle images have variability due to variation in the orientation of the particles in sample and also have variability due to conformational heterogeneity between structures. To be able to identify the particles and

their conformations, semantic representations should be invariant to these transformations. Here, we demonstrate the capacity of TARGET-VAE to learning different classes and conformations of particles in cryo-EM micrographs.

**Identifying different classes of particles in EMPIAR-10025.**   The dataset EMPIAR-10025 [34] contains T20S proteasome micrographs used to solve the T20S proteasome structure at 2.8 Å resolution. We use the Topaz [35] to create a stack of 161,292 400x400 images of T20S proteasome particles. After downsampling by factor of 4 and normalizing the images, we train a $P_8$ TARGET-VAE with z_dim=2. Due to the size and variability of these images, we use a larger spatial generator network than the one used on the MNIST datasets. Here, the generator has six fully-connected layers where each layer has 512 hidden units. We use a uniform prior over $r$. We randomly select 10% of the images as the validation set to monitor the training process. After training, we apply the inference model on the images of the particles to extract the most-probable semantic representations (same as the clustering of the MNIST datasets described in 4.3). We find that these representations produce meaningful semantic clustering of the particles, grouping them by view and the presence of contaminants (gold particles in the dataset), and also can be used to reconstruct the particles with specified rotation and translation (Figure 4).

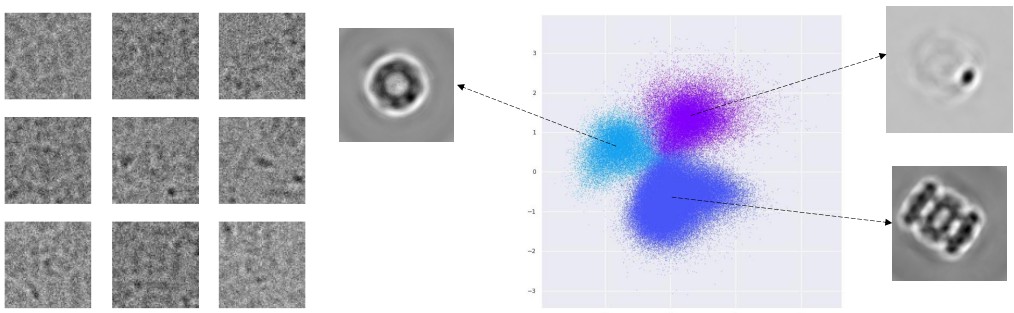

Figure 4: Left: Images of T20S proteasome particles from EMPIAR-10025. These images are extremely noisy. Right: The 2D latent representation space for EMPIAR-10025 learned by TARGET-VAE. TARGET-VAE identifies three clusters from which we show reconstructed examples produced by the spatial generator network.

**Identifying different views of particles in EMPIAR-10029.**   EMPIAR-10029 is a simulated EM dataset of GroEL particles. This dataset has 10,000 200x200 images, which we downsample to 100x100 pixels and normalize. We train the TARGET-VAE with the same settings as for EMPIAR-10025. The latent space does not identify separate clusters in this dataset, likely because a continuum over views is present from simulation, but exploration of the learned semantic latent space identifies GroEL projections independent of their rotation and translation (Figure 5).

TARGET-VAE is also able to discover continuous latent heterogeneity in other cryo-EM datasets (Appendix A.8).

## 5   Conclusion

We present TARGET-VAE, a translation and rotation group equivariant variational autoencoder framework for learning translation and rotation invariant object representation in images without supervision. By structuring the encoder to be translation and rotation equivariant, the model learns latent rotation and translation variables, disentangling these transformations from object semantics encoded in a separately learned unstructured semantic latent variables, by training it jointly with a spatially equivariant generator network. TARGET-VAE learns to accurately predict object locations and rotations, without any supervision, and also learns content representations that reflect known semantics (i.e., clusters match known semantic labels) across multiple datasets. Although we only consider 2d images containing single objects with rotation and translation, this framework can be extended to other object transformations by adopting encoder networks with the proper equivariances and adapting the approximate posterior distribution. In the future, we expect this framework can also

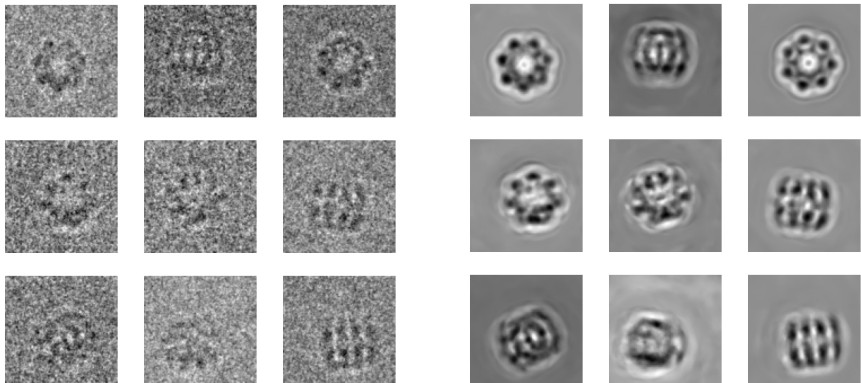

Figure 5: Left: Simulated GroEL particle images from EMPIAR-10029. Right: The reconstructed particles. TARGET-VAE learns the distribution over particle views controlled for in plane rotation and translation and generates de-noised particle views.

be extended to multi-object detection, object tracking over time, and to 3d environments. TARGET-VAE lays the groundwork for a new generation of fully unsupervised object detection and semantic analysis methods for cryo-EM and other imaging modalities.

## Acknowledgments and Disclosure of Funding

This work was supported by a grant from the Simons Foundation (SF349247).

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
