# Unsupervised Object Representation Learning using Translation and Rotation Group Equivariant VAE (Supplementary Material)

**Alireza Nasiri**
Simons Machine Learning Center
New York Structural Biology Center
anasiri@nysbc.org

**Tristan Bepler**
Simons Machine Learning Center
New York Structural Biology Center
tbepler@nysbc.org

## A   Appendix

### A.1   Calculating Kullback-Leibler divergence

Based on the standard definition for the KL-divergence, we have:

$$
\begin{aligned}
KL(q(z,\theta,t,r|y)||p(z,\theta,t,r)) &= \sum_{z,\theta,t,r} q(z,\theta,t,r|y)log\frac{q(z,\theta,t,r|y)}{p(z,\theta,t,r)} \\
&= \sum_{z,\theta,t,r} q(t,r|y)q(\theta|t,r,y)q(z|t,r,y)log\frac{q(t,r|y)q(\theta|t,r,y)q(z|t,r,y)}{p(t,r)p(\theta)p(z)} \\
&= \sum_{t,r} q(t,r|y)log\frac{q(t,r|y)}{p(t,r)} + \\
&\quad \sum_{z,\theta,t,r} q(t,r|y)q(\theta|t,r,y)q(z|t,r,y)log\frac{q(\theta|t,r,y)q(z|t,r,y)}{p(\theta)p(z)}
\end{aligned}
\tag{1}
$$

To simplify this equation, we define the first part of the result from Equation 1 as:

$$
KL_{t,r} = \sum_{t,r} q(t,r|y)log\frac{q(t,r|y)}{p(t,r)}
\tag{2}
$$

The second part of the result from Equation 1 can be further expanded as:

$$
\begin{aligned}
&\sum_{z,\theta,t,r} q(t,r|y)q(\theta|t,r,y)q(z|t,r,y)log\frac{q(\theta|t,r,y)q(z|t,r,y)}{p(\theta)p(z)} \\
&= \sum_{t,r} q(t,r|y)\left(\sum_{\theta} q(\theta|t,r,y)\,log\frac{q(\theta|t,r,y)}{p(\theta)} + \sum_{z} q(z|t,r,y)log\frac{q(z|t,r,y)}{p(z)}\right) \\
&= \sum_{t,r} q(t,r|y)\left(KL\left(q(\theta|t,r,y)||p(\theta)\right) + KL\left(q(z|t,r,y)||p(z)\right)\right)
\end{aligned}
\tag{3}
$$

Assuming these definitions for $KL_\theta$ and $KL_z$:

$$
KL_\theta = KL\left(q\left(\theta|t,r,y\right)||p\left(\theta\right)\right)
\tag{4}
$$

$$
KL_z = KL\left(q\left(z|t,r,y\right)||p\left(z\right)\right)
\tag{5}
$$

36th Conference on Neural Information Processing Systems (NeurIPS 2022).

, we can rewrite the the Equation 1 using Equations 2, 3, 4, and 5 as:

$$KL(q(z, \theta, t, r|y)||p(z, \theta, t, r)) = KL_{t,r} + \sum_{t,r} q(t, r|y)(KL_\theta + KL_z)$$ (6)

## A.2 MNIST(N) and MNIST(U) datasets

We generated two datasets of MNIST(N) and MNIST(U), by rotating and translating digits in MNIST. The rotation angles of digits for MNIST(N) are randomly sampled from $\mathcal{N}(0, \frac{\pi^2}{16})$, and for MNIST(U) are randomly sampled from $U(0, 2\pi)$ (Figure 1). Images in both of the datasets are 50x50 pixels.

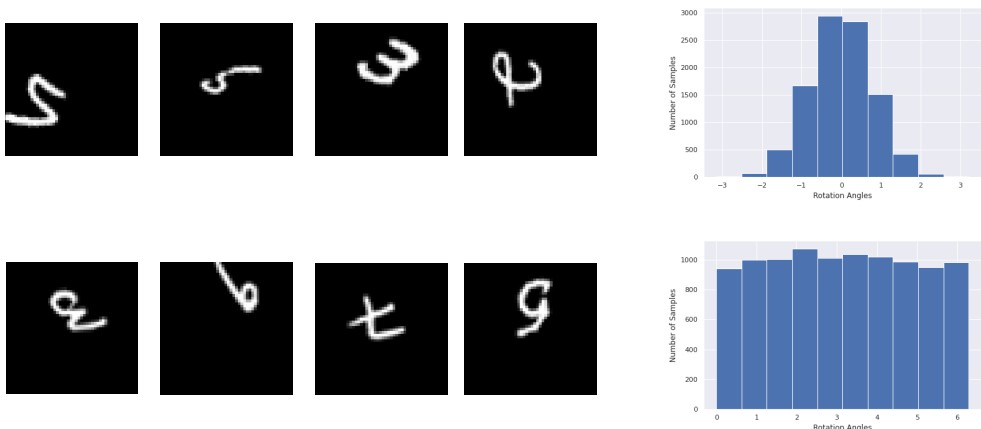

Figure 1: Left: Instances of MNIST(N) (top), and MNIST(U) (bottom) datasets. Right: Distribution of rotation angles in the test set of MNIST(N) (top), and MNIST(U) (bottom) datasets.

## A.3 Digit-wise rotation correlation, and RMSE of the predicted rotations

Following our discussion about the difference between the predicted angles and the ground-truth values, we measure the digit-wise rotation correlation for each dataset (Figure 2). We recognized that some predicted rotations are off by about $\pi$ from their ground-truth angles for digits 0, 1, and 8. We suspect that this is caused by the symmetry of these digits where for example a hand-written digit 8 with rotation $\pi$, looks roughly the same as that digit with rotation zero.

To study the accuracy of the predicted rotation angles by TARGET-VAE, we calculate the mean standard deviation of the predicted rotations, introduced in [1]. This metric basically measures the mean square error between the rotation of the object in the input image and the predicted rotation for that object. We rotate each image in the test set of MNIST(U), 160 times using angles uniformly sampled from $[0, 2\pi]$, and then pass them through the trained inference model to get the predicted rotation angle for them. Since assigning predicted rotation of zero to an object is arbitrary in our framework (model might assign predicted rotation angle of zero to the object that is actually rotated 90 degrees), we subtract the predicted rotation for the objects from the predicted rotation value when the input object is not rotated. Table 1, shows the root mean square error (RMSE) calculated in average for each digit's predicted rotation. As expected, the RMSE of rotations is highest when there is no inference done on rotations as in spatial-VAE [2], and using finer discretization in rotation inference results in more accurate prediction of the rotation values. Some digits such as 0, 1, and 8, due to their symmetry, have less accurate rotation predictions compared to the other ones.

## A.4 TARGET-VAE identifies multiple objects without supervision

We created a new dataset using multiple rotated and translated digits from MNIST(U). We call this new dataset MNIST(multi) which we created by randomly sampling rotated and translated digits from MNIST(U) and inserting them in random offsets of a 150x150 pixel image. We use the model trained on MNIST(U) to identify the translation, rotation, and the content latent for the digits in

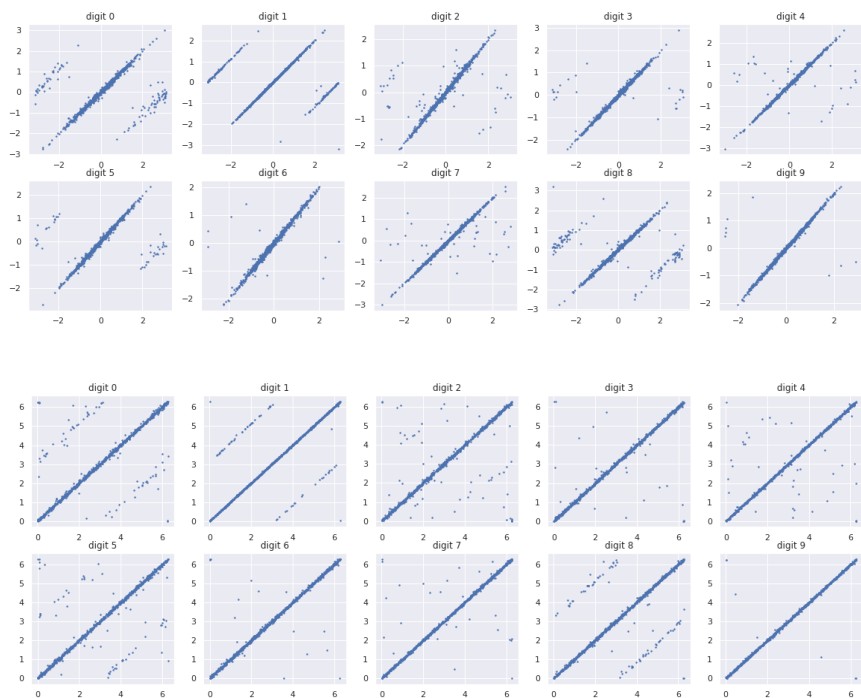

Figure 2: The predicted rotation values (x axis) and the ground-truth rotation angles (y axis) for digits in MNIST(N) (top), and MNIST(U) (bottom) using TARGET-VAE with model $P_8$ and z_dim=2. Some predicted rotations for digits 0, 1, and 8 are off by $\pi$ from their ground-truth values.

Table 1: The RMSE of the predicted rotations over MNIST(U) with z-dim=2

| Digits | 0 | 1 | 2 | 3 | 4 | 5 | 6 | 7 | 8. | 9 | AVG |
|---|---|---|---|---|---|---|---|---|---|---|---|
| Spatial-VAE [2] | 96.50 | 98.73 | 97.83 | 97.88 | 98.59 | 98.06 | 97.82 | 98.10 | 97.62 | 98.55 | 97.97 |
| TARGET-VAE $P_4$ | 31.57 | 33.86 | 32.36 | 11.66 | 27.08 | 20.94 | 6.26 | 8.15 | 28.95 | 6.68 | 20.75 |
| TARGET-VAE $P_8$ | 32.81 | 22.04 | 16.69 | 6.36 | 20.71 | 16.24 | 13.00 | 15.84 | 16.97 | 7.80 | 16.85 |
| TARGET-VAE $P_{16}$ | 17.39 | 18.47 | 14.74 | 5.79 | 7.65 | 12.09 | 3.40 | 4.63 | 17.75 | 2.74 | 10.47 |

MNIST(multi). We find that the model correctly identifies and reconstructs the objects (Figure 3). Even though the model, can identify non-overlapping objects in this approach, it struggles to identify the overlapping objects. We believe these preliminary results show the potential of our model to be used in multiple objects detection.

## A.5 TARGET-VAE predicts the rotation of the shapes in dSprites and clusters them with high accuracy

The dSprites dataset was introduced to benchmark unsupervised disentangled representation learning methods [3]. It contains 64x64 images of three shapes: squares, ellipses, and hearts. Each shape is rotated by one of 40 values linearly spaced in $[0, 2\pi]$, translated across both $x$ and $y$ dimensions, and scaled using one of six linearly spaced values in [0.5, 1]. We train TARGET-VAE with $P_8$ group convolution and z_dim=2 on dSprites, using a uniform prior over $r$ and $\mathcal{N}(0, \pi)$ prior over $\theta$. TARGET-VAE learns content representations that capture the type and scale of the objects, but is invariant from their location and rotation (Figure 4). Furthermore, TARGET-VAE accurately predicts the rotation of the shapes, and, interestingly, the symmetry groups of the shapes become immediately apparent when examining the correlation between the predicted and ground truth rotations.

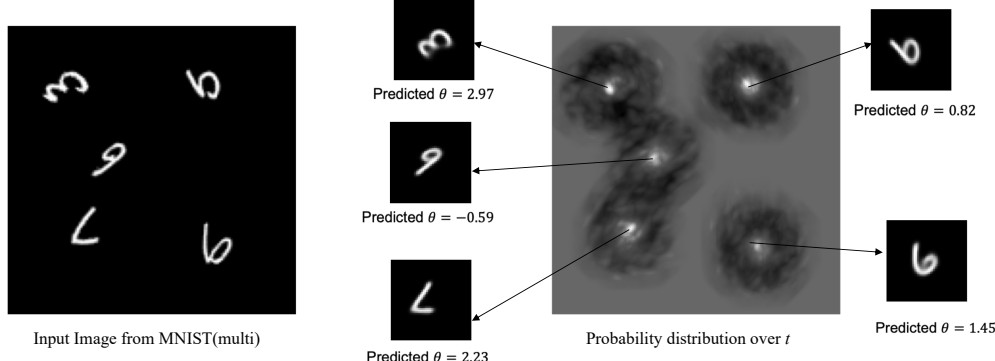

Predicted $\theta = 2.97$

Predicted $\theta = 0.82$

Predicted $\theta = -0.59$

Input Image from MNIST(multi)

Predicted $\theta = 2.23$

Probability distribution over $t$

Predicted $\theta = 1.45$

Figure 3: Left: Input image with multiple rotated and translated digits from MNIST(multi); Right: Probability distribution over $t$ ($q(t|y)$), the reconstructed objects and their predicted rotation angles. We marginalized $q(t, r|y)$ over $r$ to obtain $q(t|y)$ for visualization purposes. The high probability values in this attention map show the predicted locations of the objects. We use the peaks in $q(t, r|y)$ to sample from $q(\theta|t, r, y)$ and $q(z|t, r, y)$, the predicted rotation and content values for each object. The sampled $\theta$, $t$, and $z$ values are used to reconstruct each individual object.

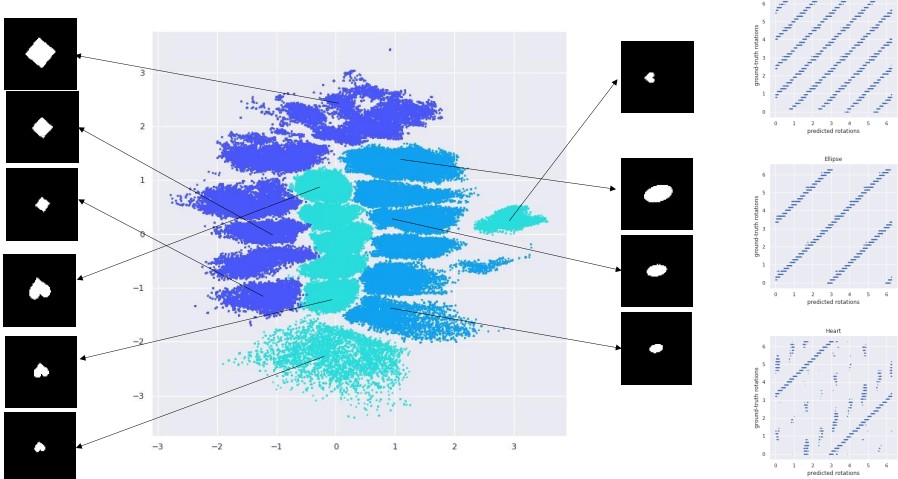

Figure 4: Left: Two-dimension latent space of dSprites using TARGET-VAE with $P_8$. Right: Correlation between rotations given by TARGET-VAE and ground truth rotations for squares, ellipses, and hearts.

### A.6 TARGET-VAE learns transformation-invariant representations in the galaxy zoo dataset

Galaxy zoo contains images of galaxies gathered by Sloan Digital Sky Survey [4]. This dataset contains more than 61,000 RGB images, which we cropped and downsampled to 64x64 pixels. The galaxies appear with different rotations and translations in the images. We train TARGET-VAE with $P_8$ group convolution on the train set. TARGET-VAE learns to accurately predict the rotation and translation of the galaxies and reconstructs centered and aligned galaxy images when generating with $t$ and $\theta$ set to zero (Figure 5).

### A.7 Ablation studies

We conducted extensive ablation study to validate the effectiveness of the main components of our proposed method.

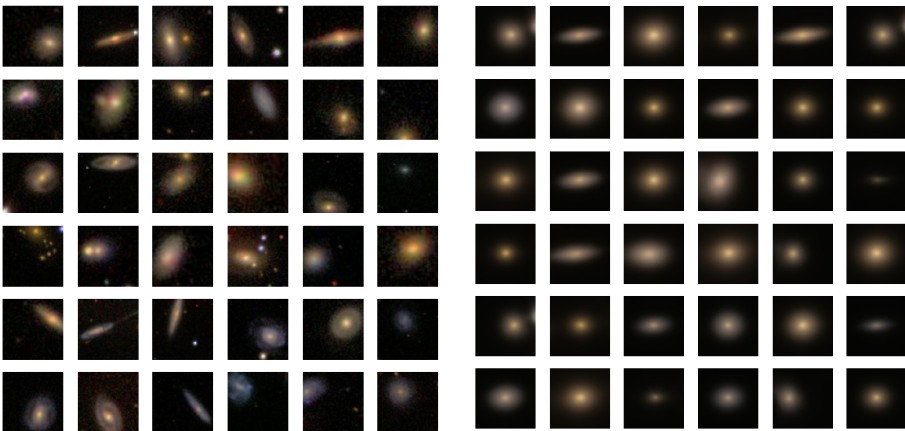

Figure 5: Left: Images of galaxy dataset used for testing. Right: Reconstructed images by TARGET-VAE with $P_8$ and z_dim=2 and using the transformation-invariant representations, where rotation and translation are set to 0.

**Variant 1 - Inference only on translation**    We evaluated the importance of the rotation equivariant features, by modifying the inference model to use regular convolutional layers instead of the group convolutional ones. As a result, the posterior distributions will be $q(t|y)$, $q(z|t, y)$, and $q(\theta|t, y)$, where they only depend on the input $y$ and the translation value $t$. We observed that, as expected, eliminating inference on the discretized rotation dimension has a significant negative effect on identifying transformation-invariant representations and the clustering accuracy on MNIST(U) is only 33.8% (Table 2).

**Variant 2 - Inference only on translation + using group convolutional layers**    In this variant, we use the group convolutional layers in the inference model, but we do not perform any inference on the rotation dimension. We apply a fully-connected layer on the output of the group convolutional layers to map the rotation values at every location to a single value. As a result, the posterior distribution are the same as variant 1 and they do not depend on $r$. We experiment with this variant to identify how effective is the use of the group convolution layers. It turns out that if we do not perform any inference on the rotation dimension, just using the group convolutional layers for feature extraction, only slightly improves the clustering accuracy.

**Variant 3 - Inference on both translation and rotation without adding $\theta_{offset}$:**    In this variant, we use group convolutional layers and we perform inference on both translation and rotation. The difference between this variant and our proposed framework is that we are not adding $\theta_{offset}$ of kernels in each rotation dimension to the $q(\theta|t, r, y)$. Adding $\theta_{offset}$ to the posterior on $\theta$, allows us to break down the rotation space among the $r$ discretized rotations, and without it, the model proves not capable of identifying the rotation of the digits in the MNIST(U) dataset.

**Variants 4 to 6 - Increasing the level of discretization of the rotation space**    In these variants, we perform inference on both rotation and translation, and we add $\theta_{offset}$ of each rotation dimension to the mean of its corresponding q($\theta$|t,r,y) distribution. In these variants, we show that by increasing the level of discretization of the rotation space, model can have a better estimate of the actual rotation values, and this in turn helps the model with improving the clustering accuracy.

Table 2, shows the translation and rotation correlation, along with the clustering accuracy of the mentioned variants on the MNIST(U) dataset. Performing inference on rotation plus adding $\theta_{offset}$ to the posterior on $\theta$, significantly increases the rotation correlation and the clustering accuracy.

Table 2: Performace of variants in the ablation study on MNIST(U)

| Model | Group Conv | Translation Corr | Rotation Corr | Clustering Accuracy |
|-------|-----------|------------------|---------------|---------------------|
| Variant 1 | - | 0.966, 0.967 | 0.005 | 33.8% |
| Variant 2 | p4 | 0.967, 0.967 | 0.005 | 37.1% |
| Variant 3 | p4 | 0.968, 0.972 | 0.008 | 36.6% |
| Variant 4 | p4 | 0.975, 0.976 | 0.80 | 56.6% |
| Variant 5 | p8 | 0.972, 0.971 | 0.859 | 57.1% |
| Variant 6 | p16 | 0.974, 0.971 | 0.93 | 63.4% |

## A.8 Learning translation-rotation-invariant representations of proteins with TARGET-VAE

In cryo-EM images, rotation and translation are the major transformations that cause variations in particles in the micrograph. Here, we show the result of our experiments with TARGET-VAE to learn the translation-rotation-invariant representations, for two cryo-EM datasets.

**Identifying hinge motion of 5HDB** We train our model on a dataset of 20,000 simulated projections of integrin $\alpha$-IIb in complex with integrin $\beta$-3 (5HDB) [5]. We aim to identify the translation and rotation invariant representations of the protein to be able to identify the variations in its structure. We train TARGET-VAE with $P_8$ and uniform prior over $\theta$. Since there is less variation in the data, we set $z\_dim$ to 1. After training, we sample from the representation latent space and reconstruct the images with no rotation and translation. Figure 6 shows some examples of the reconstructed particles. We observe that the reconstructed images identify the hinge motion of the particle.

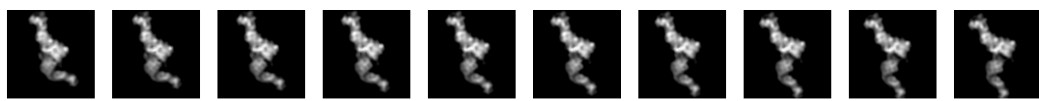

Figure 6: Reconstructed proteins from 5HDB dataset by sampling from 1D representation space, shows the movement in the lower part of the particle.

**Learning arm motion of particle in CODH/ACS** We have about 14,000 40x40 pixels images of the CODH/ACS protein complex. We train TARGET-VAE with $P_8$ and $z\_dim = 2$, for 100 epochs. The prior over $\theta$ is uniform and we set the generator to have 6 fully-connected layers with 512 hidden units in each. After training, we sample from the transformation-invariant representation space and reconstruct the particle to identify the different conformations of the protein. Figure 7 shows the movement on the upper and lower parts of the particle.

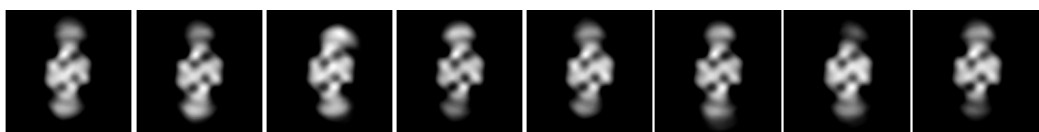

Figure 7: Reconstructed proteins from CODH/ACS dataset, where the motion in the upper and lower arms of the particle can be captured.