# OpenReview forum: "Unsupervised Object Representation Learning using Translation and Rotation Group Equivariant VAE"
_NeurIPS.cc/2022/Conference — NeurIPS 2022 Accept_

### Official Review · Reviewer_Gmwc · 2022-07-11

**Rating:** 6
**Confidence:** 4
**Soundness:** 3 good
**Presentation:** 3 good
**Contribution:** 3 good

**Summary:**

This paper presents an equivariant auto-encoder for object-centric representation. specifically, the group-equivariant encoder takes as input an image and outputs the disentangled latent representation -- an object is decomposed into pose and appearance. To learn the latent representation, the decoder reconstructs the input images in an equivariant way. In this case, the proposed method can learn the group-equivariant representation in an unsupervised style. Different from existing work, this method uses a better encoder and a better way to relate pose and appearance. The paper further validates their method in multiple datasets including a realistic application. And it shows better semantic learning and poses regression.

**Questions:**

Overall, my suggestion is to relate the work with Canonical capsules -- e.g., estimate the performance of canonicalization -- and include more details of multi-object experiments -- for example, we could analyze the detection performance, and canonicalization in the multi-object setting.

**Limitations:**

The authors addressed the limitations well.

**Strengths And Weaknesses:**

Originality:
This work shows the potential of group equivariance in unsupervised representation learning, shedding light on symmetry-enforced learning. Different from previous works that show weak results, this paper gives a reasonable performance in both synthetic and realistic datasets. The only concern I have in this aspect is the missing discussion with closely related Canonical Capsules [1]. While the canonical capsule is proposed for 3D learning, it shares a similar target -- decomposing the object with disentangled factors. More importantly, both method shows the effectiveness of this decomposed representation, pose learning. It might be interesting to relate these two works.

[1], Sun et al. Self-supervised capsules in canonical pose. NeurIPS 2021

Quality:
The work validates their method with multiple datasets. The claims are well-supported.
- One concern I had is the missing results in the commonly used benchmark for the unsupervised object discovery (the dataset used in Slot attention). In this case, we assume the objects are aligned. This might need to be justified.
- And, to better understand the proposed method, one more suggestion I have is to perform the estimation of canonicalization as in the canonical capsule paper. When disentangling objects into pose and appearance, we already factor out the pose -- in some sense, we model the appearance of objects in the canonical pose.
- I think multi-object experiments might be more interesting than single-object experiments. But the details of multi-object experiments are omitted. It would be cool to show more metrics like the performance of unsupervised detection.

Clarity:
The paper is overall easy to follow. From my viewpoint, the only concern is about the detail of the encoder. I think it might be more clear if the author can provide more implementation details regarding how to generate the different factors.

Significance:
I think the results shown in this paper are interesting to our research community. It validates the effectiveness of group-equivariant representation in an unsupervised setting. In particular, the paper shows good results on multi-object images. This enables the group-equivariant to deal with the scene dataset. So I strongly recommend having more discussion in the multi-object experiments.

---

> ### Author Response · Authors · 2022-08-02
> **Authors' Response to Reviewer Gmwc**
>
> We thank the reviewer for taking the time to review our work and for the constructive remarks and questions. We have now edited the manuscript to address these questions and otherwise clarify the model presentation. Detailed responses to the reviewer’s comments and questions follow below.
>
> > One concern I had is the missing results in the commonly used benchmark for the unsupervised object discovery (the dataset used in Slot attention). In this case, we assume the objects are aligned. This might need to be justified.
>
> In this work, we are focused on disentangling the rotation and translation from the object representation. To this end, we benchmarked our method on the datasets that are commonly used for this purpose such as dSprites [1], transformed MNIST, and electron-microscopy images. The datasets mentioned by the reviewer are multi-object datasets which are not the focus of this current paper (though we are excited about this direction for future work!).
>
> >And, to better understand the proposed method, one more suggestion I have is to perform the estimation of canonicalization as in the canonical capsule paper. When disentangling objects into pose and appearance, we already factor out the pose -- in some sense, we model the appearance of objects in the canonical pose.
>
> Thanks for this suggestion. One of the advantages of our equivariant encoder is that canonicalization is guaranteed up to the rotation group. We now include canonicalization performance calculated according to the metric from [2] in the mansucript. The results agree with our rotation prediction correlation results, as shown in appendix A3, Table 3. Further discussion about this experiment is added to the paper.
>
> >I think multi-object experiments might be more interesting than single-object experiments. But the details of multi-object experiments are omitted. It would be cool to show more metrics like the performance of unsupervised detection.
>
> We are also excited about the extension of our framework to multi-object detection problems. We performed some preliminary experiment, where we trained the model on single object images, and ran evaluation on multi-object MNIST dataset, which we included its results in Appendix A4. This current approach has the limitation of not being able to correctly identify the overlapping objects, but we think a full extension of TARGET-VAE to multi-object detection, is an exciting next step.
>
> >The paper is overall easy to follow. From my viewpoint, the only concern is about the detail of the encoder. I think it might be more clear if the author can provide more implementation details regarding how to generate the different factors.
>
> We further clarified the training process and the related details by revising Fig. 1, and modifying the text related to the model’s training process.
>
> [1] Matthey et al. dSprites: Disentanglement testing Sprites dataset, 2017.
>
> [2] Sun et al. Self-supervised capsules in canonical pose. NeurIPS 2021.

---

### Official Review · Reviewer_2xXA · 2022-07-13

**Rating:** 4
**Confidence:** 3
**Soundness:** 2 fair
**Presentation:** 2 fair
**Contribution:** 2 fair

**Summary:**

This paper proposes a model called Translation and Rotation Group Equivariant VAE (TARGET-VAE) which the authors claim can learn semantic object representations that are invariant to pose and location by structurally decomposing the image generative factors into the semantic, rotation, and translation components. Experiments are conducted on MNIST and a microscopy protein dataset to show that it can estimate the rotation and translation accurately.

**Questions:**

See above

**Limitations:**

Unfortunately, the authors did not include any limitations and potential negative societal impact discussion.

**Strengths And Weaknesses:**

Pros:

The use of group equivariant convolution makes sense to me, and the results show that it can model rotation better than spatial VAE indeed.

Cons:

This paper is somewhat overclaiming and the experiment section is weak. The experiments are only conducted on MNIST and a protein dataset called EMPIAR-10025 with very few baselines. I'm not familiar with the protein dataset, but from the illustration in Fig.5, I don't think it could be called unsupervised "object" or "semantic" representation learning when only verifying on these simple datasets. If the authors genuinely believe their method is general and effective, I suggest they try it on more general RGB data like ImageNet, or at least CIFAR.

Fig.1 is unclear, making it hard to understand without referring to the text. I suggest the authors add more context to the figure and caption to make it more self-contained and help readers understand it better.

The motivation is not well addressed as the authors did not motivate why disentangling rotation and translation are essential for the generative models (maybe it is well known in the bioinformatics field but it is not reflected in this paper). The results also did not show the generated results but only the predicted translation/rotation results (I think a vanilla spatial transformer network should also be able to do that).

---

> ### Author Response · Authors · 2022-08-02
> **Authors' Response to Reviewer 2xXA**
>
> Thank you for taking the time to review our work and provide feedback. We respond to the reviewer’s specific comments below.
>
> >This paper is somewhat overclaiming and the experiment section is weak. The experiments are only conducted on MNIST and a protein dataset called EMPIAR-10025 with very few baselines. I'm not familiar with the protein dataset, but from the illustration in Fig.5, I don't think it could be called unsupervised "object" or "semantic" representation learning when only verifying on these simple datasets. If the authors genuinely believe their method is general and effective, I suggest they try it on more general RGB data like ImageNet, or at least CIFAR.
>
> In our extensive experiments, we show that our model learns to accurately predict the rotation and translation of objects (based on our experiments on two transformed MNIST datasets (Results 4.2, Appendix A.3), dSprites (Appendix A.5), four cryo-EM datasets (Results 4.4, Appendix A.8), and galaxy zoo (Appendix A.6)) without supervision, and that the object representations learned are semantically rich; we are able to accurately  cluster semantically similar objects despite their varying rotation and translation (Results 4.3 and 4.4, Appendix A.5). Furthermore, we demonstrate a preliminary application of this model to unsupervised multi-object detection (Appendix A.4). We perform an extensive ablation study of our model to demonstrate the criticality of our group conv-based encoder and mixture model approximate posterior distribution, as well as examining the impact of varying fine-grained-ness of the group convolution on performance (Results 4.2 and 4.3, Appendix A.7). We also demonstrate significant improvement over previous work on this unsupervised learning problem. Most crucially, we apply our model to real life microscopy datasets and demonstrate its ability to discover real semantic variation and cluster proteins in cryoEM datasets despite the random orientations and translations of the proteins. We show some results on an RGB dataset in the Appendix, but natural images have many other sources of variation unrelated to the primary object and its pose, such as background, which we do not address in this work. Extending our model to explicitly separated foreground objects from background could be interesting to explore in the future, but it is outside the scope of this work, and is not relevant to our primary application of interest, electron microscopy.
>
> >Fig.1 is unclear, making it hard to understand without referring to the text. I suggest the authors add more context to the figure and caption to make it more self-contained and help readers understand it better.
>
> We modified Fig. 1, and added more details to the figure and its caption to better clarify the overall method.
>
> >The motivation is not well addressed as the authors did not motivate why disentangling rotation and translation are essential for the generative models (maybe it is well known in the bioinformatics field but it is not reflected in this paper).
>
> To motivate this work, we have described examples from satellite imagery, natural images, and cryo-electron microscopy. Particularly in cryo-electron microscopy, disentangling rotation and translation is important for generative models because it otherwise obscures the structure of the particle and impedes 2D analysis and 3D reconstruction. Rotations and translations are also problematic for image registration. We expand on this in the manuscript.
>
>
> > The results also did not show the generated results but only the predicted translation/rotation results (I think a vanilla spatial transformer network should also be able to do that).
>
> Examples of generated images can be found in Figures 4 and 5, in the main paper, and in Appendix Figures 3, 4, 5, 6, and 7. Spatial transformer networks are neither unsupervised nor structurally equivariant to these transformations, unlike our model.

---

> > ### Comment · Reviewer_2xXA · 2022-08-09
> > **Response to authors**
> >
> > As I'm not familiar with bioinfomatics and electron microscopy, I admit that it is hard for me to evaluate the impact of the proposed model. As a researcher in the computer vision community, I feel like the performance is somewhat limited (for example, the reconstruction results in A.6 are very blurred and lost many details). Results in Fig.4, 5 and A6 are for reconstruction and I don't see new synthetic data, which is VAE used for. I believe that the rotation/translation prediction can also be done with a spatial transformer network w/ VAE or even AE/Denosing AE as the STN learning is implicit and does not require explicit gt, which should fit the authors' context.
> >
> > I decide to keep my rating as borderline reject.

---

### Official Review · Reviewer_Uca2 · 2022-07-18

**Rating:** 6
**Confidence:** 4
**Soundness:** 3 good
**Presentation:** 3 good
**Contribution:** 3 good

**Summary:**

The paper proposes TARGET-VAE, a representation learning method that disentangles the appearance, position, and rotation of an object in a given image. While the model is applied mainly to single-object images, it also provides some preliminary results that suggest that extending to multi-object images should be possible.

The method is basically a VAE with some inductive biases added to both encoder and the decoder. The inductive bias added to the encoder is what mainly distinguishes it from its main baseline -- Spatial-VAE. The central claim of this paper is that: using a group convolution network in the encoder leads to significant performance gains in disentangling the object rotation. Their empirical results show this to be true.

The encoder is basically a group convolution network -- which takes an image and applies a filter by both shifting and rotating the filter over the image. This gives a feature vector/cell for every possible shift and rotation. Each cell computes a logit which is used to select or activate the cell at a specific shift and rotation (using Gumbel-Softmax). The activated cell is then used to infer the latents for appearance, position, and rotation of the object. The decoder is directly taken from Spatial-VAE. It applies an explicit affine transformation to the object glimpse based on the given rotation and position.



**Questions:**

1. After doing Gumbel-Softmax in L180, how is the soft one-hot vector used? Is it used to weighted-sum the features of the G-feature map? Also, how is softmax applied on q(t,r|y) (in L223 and L245)? Is this followed by an arg max?

2. The details of the prior applied on rotation $\theta$ is unclear.

3. In L180, is the q(t, r) a typo where the intention was to write q(t, r|y)?

4. If my understanding is correct, then it might prevent confusion by saying $\theta_{offset}$ 'belongs' rather than 'equal' to the set of angles in L137.

5. Why not have a $\Delta x$ for position similar to $\Delta \theta$ for rotation? It seems even the position can also suffer from discretization errors (just like rotation). Then, why not allow the position to also be freely adjusted/corrected using the selected $(t,r)$ feature. Also, another related question is: what is $x$ in L138? The confusion arises because (t, r, theta, z) are the only latents that are mentioned in equation (6).

**Limitations:**

Yes, the paper discusses the limitations and points to future directions such as possible extensions to other transformations (beyond rotation and position), multi-object scenes, and 3D scenes.

**Strengths And Weaknesses:**

Strengths:
1. The paper proposes a novel encoder design to help disentanglement of object rotation.
2. Experiments focusing on single-object images show that the proposed model can infer rotation quite accurately while the baseline suffers significantly. The results also show that having a separate latent to represent the rotation relieves the burden on the appearance latent -- their results on clustering the appearance latent of MNIST digits show better clustering performance than the baselines.
3. Paper does meaningful qualitative analysis of the model.
4. Preliminary results suggest that extending to multi-object images should be possible. This may be of interest to the object-centric learning community where previous works have not explicitly decomposed the rotation angle. I feel perhaps this result could have been highlighted more in the main paper.
5. Design seems extensible to some new factors of variation (not just position and rotation) by introducing an appropriate group convolution.

Weaknesses:
1. The disentanglement is still driven by the fact that the decoder performs an explicit rotation transformation of the canonical image by taking the rotation angle as input. This seems to suggest that when extending the model to other factors of variations such as color, one may need a decoder that takes the explicit color value as input -- this can become challenging very quickly when scaling to more such factors of variation.
2. The model design and its presentation in the paper could be made cleaner. (see the Questions section for more details)
3. It is a bit unjustified to say that the method 'detects and represents objects' without supervision because the model was trained on images that explicitly have one object.

---

> ### Author Response · Authors · 2022-08-02
> **Authors' Response to Reviewer Uca2**
>
> We thank the reviewer for taking the time to review our work and for the constructive remarks and questions. We have now edited the manuscript to address these questions and otherwise clarify the model presentation. Detailed responses to the reviewer’s comments and questions follow below.
>
> >The disentanglement is still driven by the fact that the decoder performs an explicit rotation transformation of the canonical image by taking the rotation angle as input. This seems to suggest that when extending the model to other factors of variations such as color, one may need a decoder that takes the explicit color value as input
>
> This is a good point and certainly true. One advantage of the structure of our decoder is that it naturally extends to any transformation that can be encoded as transformations of the coordinate space, but these extra sources of variation, however parameterized, would need to be accepted explicitly as input in order to generate the correct coordinate transformation. On the one hand, this is a limitation on the number and kinds of variation that can be encoded, but, on the other hand, it guarantees that these sources of variation are well understood and explicitly controlled. In contrast, if these were encoded through unstructured variables, there would be no guarantee that those variables would always correspond to any particular transformation.
>
> >It is a bit unjustified to say that the method 'detects and represents objects' without supervision because the model was trained on images that explicitly have one object.
>
> It is true that the training images explicitly contain one object, but the location, orientation, and identity of those objects are unknown. In this sense, the model is completely unsupervised. It learns to perform accurate inference on the location and rotation of the object without receiving any supervision for these variables and also learns representations that identify object classes without supervision. Our method is a purely generative model, learning only from p(x), and thus is an unsupervised method in the truest sense of the term. Perhaps the reviewer specifically disliked the word “detects,” by which we meant “locates” rather than identifies the presence/absence of, though the model is able to tell where the object is not, so both meanings are generally true.
>
>
>
> And to answer to the reviewer’s specific questions:
> 1. The soft one-hot output of the Gumbel-Softmax is used to calculate the weighted sum for the parameters that describe the approximate posterior distributions over content (z), and rotation ($\theta$). It directly describes a sample from the approximate posterior over the translation, which we get by calculating the weighted sum over the translation grid points. In the evaluation step, we use arg-max to identify the most likely (t, r) and use that to identify z and $\theta$. Because we only use the maximum likelihood values of (t, r) for evaluation, softmax is not necessary (the argmax of the unnormalized (t, r) log likelihoods and the normalized ones is the same). Thank you for pointing this out, we now clarify these points in the paper.
>
> 2.  We apply a mixture prior on $\theta$ designed to approximate either a uniform prior or Gaussian prior. Because $\theta$ is defined as a mixture distribution, $p(\theta) = \sum_r p(\theta | r)p(r)$, we define priors on both $\theta|r$ and r. The prior on $\theta|r$ is a Gaussian distribution with mean given by the angle offset defined by r and standard deviation which depends on the number of discrete rotation groups, ($\frac{\pi}{r}$). For the prior on r, we usually use a uniform distribution, but a discretized normal distribution can also be used to bias the model towards certain rotation angles if a non-uniform rotation distribution is known a priori. This is now clarified in the text.
>
> 3. Thanks for pointing this out. It should be q(t, r | y). We have now corrected this in the paper.
>
> 4. We have updated the text accordingly.
>
> 5. Our method can be extended to address translations with sub-pixel values by including an additional distribution over $\Delta x$ conditioned on t as the reviewer mentions. However, because pixels already offer a fine grained grid of translation values, we decided that including this extra layer of complexity was unlikely to yield sufficiently better results to be worth it. That said, this extension would certainly be interesting to explore in the future, especially on datasets where sub pixel level translations are of interest, which could be useful for high resolution reconstruction in cryoEM, for example. To be fair, we do suspect that the small drop in translation prediction performance between our model and prior work (Spatial-VAE) for translation prediction on MNIST is likely due to our inability to learn sub-pixel level translations. . The x in L138, should be t, this is a typo, thanks for pointing this out.

---

> > ### Comment · Reviewer_Uca2 · 2022-08-09
> > **Reply**
> >
> > Thank you for your response! Most of my concerns are addressed. I maintain my score and continue to support accepting the paper.
> >
> > 1. Yes, I agree that there is a benefit to having explicit factors grounded as real-world quantities that humans can easily interpret (e.g. rotation/scale).
> > 2. The presentation was improved so I am fine with it now. Also, thank you for providing clarifications for my presentation-related questions.
> > 3. On 'the method detects and represents objects': The method does seem extensible to multi-object scenes although it is something that future work can deal with.

---

### Meta-Review · Area_Chair_Hu6Q · 2022-08-29

**Recommendation:** Accept
**Confidence:** Certain

**Metareview:**

The authors propose an effective method for group-equivariant representation learning in an unsupervised setting. The authors use group convolutional encoder in the VAE setting. This is an interesting problem in the community. The proposed method makes sense and seems technically sound. The experiment results are good and convincing in general. As pointed by multiple reviewers, however it would have been better to focus more on multi-object settings. Most of the concerns raised by the reviewers are well addressed. Although reviewer 2xXA gave relatively lower score 4, the reviewer admitted his/her lack of related background knowledge and the confidence score was low.

**Award:**

No

---

### Decision · Program_Chairs · 2022-09-14

Accept